# Crosslinking Strategies for the Microfluidic Production of Microgels

**DOI:** 10.3390/molecules26123752

**Published:** 2021-06-20

**Authors:** Minjun Chen, Guido Bolognesi, Goran T. Vladisavljević

**Affiliations:** Department of Chemical Engineering, Loughborough University, Loughborough LE11 3TU, UK; M.Chen2@lboro.ac.uk (M.C.); G.Bolognesi@lboro.ac.uk (G.B.)

**Keywords:** microgel, Janus particle, ionotropic gelation, crosslinking, cell encapsulation, enzymatic crosslinking, Photopolymerization, hierarchical microgels, composite microgels, microfluidics

## Abstract

This article provides a systematic review of the crosslinking strategies used to produce microgel particles in microfluidic chips. Various ionic crosslinking methods for the gelation of charged polymers are discussed, including external gelation via crosslinkers dissolved or dispersed in the oil phase; internal gelation methods using crosslinkers added to the dispersed phase in their non-active forms, such as chelating agents, photo-acid generators, sparingly soluble or slowly hydrolyzing compounds, and methods involving competitive ligand exchange; rapid mixing of polymer and crosslinking streams; and merging polymer and crosslinker droplets. Covalent crosslinking methods using enzymatic oxidation of modified biopolymers, photo-polymerization of crosslinkable monomers or polymers, and thiol-ene “click” reactions are also discussed, as well as methods based on the sol−gel transitions of stimuli responsive polymers triggered by pH or temperature change. In addition to homogeneous microgel particles, the production of structurally heterogeneous particles such as composite hydrogel particles entrapping droplet interface bilayers, core−shell particles, organoids, and Janus particles are also discussed. Microfluidics offers the ability to precisely tune the chemical composition, size, shape, surface morphology, and internal structure of microgels by bringing multiple fluid streams in contact in a highly controlled fashion using versatile channel geometries and flow configurations, and allowing for controlled crosslinking.

## 1. Introduction

Hydrogels are three-dimensional networks of hydrophilic crosslinked polymers that can hold large amounts of water in their intermolecular space, but they are not soluble in water in their crosslinked form. Hydrogels are widely used as excipients for drug delivery systems [1,2]; scaffolds in tissue engineering [3,4]; wound dressings [5]; absorbents for hygiene products (diapers, napkins, hospital bed sheets, and sanitary towels) [6]; gelling agents, thickeners, and packaging materials in food products [7]; and irritation-free, transparent materials for contact lenses [8]. Their high-water retention capacity and soft, porous structure mimic the in vivo extracellular matrix (ECM) microenvironment. Hydrogels can undergo large and reversible volume changes by expelling or absorbing water in response to external stimuli (light, temperature, pH, ionic strength, and chemical triggers), which can dramatically change their physical properties and permeability to small molecules [9].

In “physical” hydrogels, molecular entanglements and/or secondary forces such as ionic, H-bonding, or hydrophobic forces play a main role in the network formation [10]. Physical gels are reversible and can be disintegrated by changing environmental conditions, such as the pH, temperature, and ionic strength of the solution. Typical physical hydrogels, such as alginate, carboxymethyl cellulose, and chitosan, are prepared by ionotropic gelation with oppositely charged divalent ions [11,12]. In “chemical” gels, polymer chains are permanently connected by covalent bonds. Chemical gels are prepared in two different ways, namely: (i) free radical polymerization of low molecular weight hydrophilic monomers and (ii) polymerization of soluble polymers. Free-radical polymerization often results in a significant level of residual monomers, and therefore, hydrogels must be purified to remove unreacted monomers, which are often harmful.

Microgels are micron-sized hydrogel microparticles. They can be generated using a variety of top-down and bottom-up fabrication methods, including ultrasonication, mechanical agitation, high-pressure homogenization [13], atomization [14], extrusion through a syringe or nozzle [15], micromolding [16], and molecular self-association [17]. Compared with most of these techniques, microfluidic platforms offer superior control over the size and morphology of microgels, cell co-culture with a precise control over the number of cells of each type per single bead [18], high encapsulation efficiency due to low shear forces during droplet generation, integration of particle generation and manipulation within a single chip, continuous processing, and operation under sterile conditions. Monodisperse microgels produced in microfluidic devices have been widely used in molecular and synthetic biology [19], biotechnology, and tissue engineering [20].

Microgels can have a homogeneous (matrix-type) or core−shell structure. Matrix-type microgels are commonly synthesized via water-in-oil (W/O) emulsions by crosslinking polymers within aqueous template droplets. The incorporation of an additional phase within the dispersed phase results in the formation of multiple emulsion droplets, such as oil-in-water-in-oil (O_1_/W/O_2_), water-in-water-in-oil (W_1_/W_2_/O), or even water-in-water-in-water (W_1_/W_2_/W_3_), which can be transformed into core−shell particles by the gelation of middle phase or both inner and middle phase. The objective of this paper is to review recent developments in crosslinking polymers or monomers in microfluidic chips for the purpose of producing both matrix-type and core−shell microgels. The main emphasis was placed on the production of microgels with a complex morphology and chemical structure. The mechanisms of droplet formation in microfluidic devices have been described in many excellent review papers [21,22,23,24,25] and will not be discussed here.

## 2. Microfluidic Production of Spherical Matrix-Type Microgels

Spherical matrix-type microgels are commonly prepared using a two-step process consisting of the formation of W/O emulsion droplets containing a gel-forming polymer solution (step 1) and the crosslinking of polymer chains within the droplets (step 2). These two steps must be spatially and temporally separated, i.e., droplets must be pinched off prior to the crosslinking reaction, so as to avoid the blockage of the droplet forming channel by the gellified polymer. The crosslinking step can be completed on the chip downstream of the droplet generator or in a collection vial (off-chip).

Physical microgels are usually formed by ionic crosslinking of polymers within droplets, while chemical microgels are formed either by covalent crosslinking of polymers or by polymerizing monomers within droplets.

### 2.1. Ionic Crosslinking of Droplets in Microfluidic Channels

Ionic crosslinking occurs between charged polymer chains and oppositely charged divalent or multivalent ions. The typical example of ionic crosslinking is the electrostatic interaction between negatively charged alginate chains and positively charged Ca^2+^ ions. Ionic crosslinking in microfluidic channels can be achieved using five different strategies [26]: internal gelation, external gelation, rapid in-droplet mixing (chaotic advection), droplet merging (coalescence), and competitive ligand exchange.

#### 2.1.1. Internal (In-Situ) Gelation

Here, the dispersed phase is an aqueous solution of charged polymer containing an undissociated calcium or barium compound (shown as solid black circles in Figure 1), such as CaCO_3_, BaCO_3_ [27], tricalcium citrate [28], and Ca-EDTA [29].

The continuous phase is a mixture of hydrophobic surfactant and organic acid, usually acetic acid (CH_3_COOH), dissolved in an inert oil. The typical inert oils used in this process are vegetable oils [30,31,32,33], dimethyl carbonate [34], fluorocarbon oils [35], and hexadecane [36]. After droplet generation, the organic acid diffuses through the oil phase and dissociates at the droplet surface into acetate anions (CH_3_COO^−^) and protons (H^+^). H^+^ reacts with undissociated Ca or Ba compounds inside the droplets and releases Ca^2+^ or Ba^2+^ ions, which crosslink the polymer. In the case of the CaCO_3_/alginate/acetic acid system, the following reactions occur:(1)$${\mathrm{CH}}_{3}\mathrm{COOH}⇆{\mathrm{CH}}_{3}{\mathrm{COO}}^{-}+H^{+}$$
(2)$$2H^{+}+{\mathrm{CaCO}}_{3}\to {\;\mathrm{Ca}}^{2+}+H_{2}O+{\mathrm{CO}}_{2}$$
(3)$${\mathrm{Ca}}^{2+}+2{\mathrm{Na}}^{+}{\mathrm{Alg}}^{-}\to {\;\mathrm{Ca}}^{2+}{\left({\mathrm{Alg}}^{-}\right)}_{2}+2{\mathrm{Na}}^{+}$$

Although the method offers homogenous gelation throughout the entire droplet volume, it results in a reduction of pH below the physiological pH, which can be detrimental to cell viability [37]. To minimize the exposure of cells to a low pH, a stoichiometric amount of acetic acid can be added off-chip after droplet collection [38].

Figure 2 and Figure 3 show the microfluidic internal gelation strategies implemented to produce homogeneous alginate beads and alginate/pectin Janus beads, respectively.

Internal gelation can also be triggered by adding a slow hydrolyzing acid or a photo-acid generator (PAG) into the polymer solution rather than supplying acid from the oil phase across the droplet boundary. Morimoto et al. [39] added glucono-1,5-lactone (GDL), a slow-release acidifier to the dispersed phase, which hydrolyzed to gluconic acid to release Ca^2+^ from CaCO_3_ and to initiate alginate crosslinking without any external trigger. Photo-acid generators (PAGs) are compounds that release protons irreversibly upon illumination. Liu et al. [40] added diphenyliodonium nitrate (DPIN) to the aqueous phase to initiate the gelation of an alginate solution from the middle phase of the O/W/O emulsion droplets upon UV irradiation. The photolysis of DPIN results in the release of protons (H^+^) and hydrophobic uncharged by-products [41]:(4)$$\mathrm{DPIN}\;\overset{\mathrm{UV}}{\to }\;\mathrm{hydrophobic}\;\mathrm{by}-\mathrm{products}+H^{+}$$

To prevent these by-products from precipitating within droplets and causing channel clogging, a macrocyclic compound can be added to the dispersed phase that can bind these hydrophobic species [41]. The emulsion formulations used to produce microgels by internal gelation are summarized in Appendix A.

#### 2.1.2. External Gelation

The external gelation in microfluidic systems can be achieved using several methods, as follows. (i) An oil-soluble crosslinking agent, such as calcium acetate, can be dissolved in the oil phase and be used for on-chip crosslinking. (ii) A crosslinking agent such as CaCl_2_ can be added to the gelation bath, where the droplets are collected and crosslinked [42]. (iii) An aqueous crosslinker solution can be emulsified in a carrier oil and this emulsion can be used for on-chip crosslinking [43]. (iv) The same emulsion can be used as a shell liquid in core−shell droplets and can be used for on-chip crosslinking of aqueous cores [44]. (v) An aqueous CaCl_2_ solution can be emulsified in a carrier oil to produce a W/O emulsion, which can be dehydrated to form surfactant-coated CaCl_2_ nanoparticles dispersed in the oil phase. This nano-dispersion can be used for on-chip crosslinking [45,46]. (vi) An alcoholic CaCl_2_ solution can be dissolved in the oleic acid and can be used for on-chip crosslinking after alcohol evaporation [47,48,49]. Finally, (vii) a powdered crosslinking medium, such as dehydrated cell culture medium, can be dispersed in a carrier oil and used for on-chip crosslinking [50]. Appendix A provides examples of the microfluidic methods used for external gelation.

In the first method (i), the dispersed phase is usually an aqueous polyanion solution, while the continuous phase is an oil-soluble salt of divalent or multivalent cations, e.g., Ca^2+^, Zn^2+^, and Fe^3+^, dissolved in an oil–surfactant mixture (Figure 4). Depending of the type of oil and polymer, the oil-soluble salt can be calcium acetate [30], Fe (NO_3_)_3_ [51], CaJ_2_ [51], BaCl_2_ [52], SrCl_2_ [53], or ZnCl_2_ [54]. In the case of alginate crosslinking with calcium acetate, Ca-acetate diffuses through the oil phase to the droplet interface and dissociates in water to form Ca^2+^ ions, which trigger polymer crosslinking within the droplets, according to the following reactions:(5)$$[\mathrm{Ca}\left({\mathrm{CH}}_{3}\mathrm{COO}{)}_{2}\right]{\;}_{oil\;phase}⇆{\left[2{\mathrm{CH}}_{3}{\mathrm{COO}}^{-}+2{\mathrm{Ca}}^{2+}\right]}_{aqueous\;phase}$$
(6)$${\left[{\mathrm{Ca}}^{2+}+2{\mathrm{Na}}^{+}{\;\mathrm{Alg}}^{-}\right]}_{aqueous\;phase}\to {\left[{\mathrm{Ca}}^{2+}{\left({\mathrm{Alg}}^{-}\right)}_{2}+2{\mathrm{Na}}^{+}\right]}_{aqueous\;phase}$$

For on-chip gelation, the combined diffusion and reaction time, $\tau_{d}+\tau_{r}$, must be shorter than the residence time of the droplets in the chip, but greater than the droplet formation time, 1/*f*, where *f* is the frequency of droplet generation. The diffusive flux of salt across the oil/water interface depends on the salt concentration in the oil phase and its diffusivity in the oil phase. Another limiting factor is the salt solubility in the aqueous phase, which must be sufficiently high to trigger polymer crosslinking. The solubility of calcium salts of fatty acids in water decreases with increasing the number of carbon atoms in a molecule [30]. For example, calcium butanoate and calcium 2-ethylhexanoate cannot be used for external gelation, because no gelation occurs within 7 days when the concentration of either salt in the oil phase is 0.5 wt% [30]. The salt solubility in the oil phase could also be a limiting factor. A mixture of CaCO_3_ and acetic acid can be used instead of Ca-acetate to increase its solubility in oil [55].

The external gelation of pH sensitive Eudragit^®^ polymers can be triggered by the exchange of H^+^ ions between an organic acid dissolved in the carrier oil and the polymer dissolved in aqueous droplets. Above pH 7.4, Eudragit S100 is soluble in water because of the dissociation of carboxylic acid of methacrylic acid monomer units. A sol–gel transition occurs at pH < 7.4 because of charge neutralization (Figure 5c). Monodispersed gel beads were produced when the sol–gel transition was triggered by p-aminobenzoic acid (PABA), a weak acid with a pKa of 2.38 (Figure 5a). However, when the gelation was triggered by p-toluenesulfonic acid (PTSA), a strong acid with a pKa of −2.8, the beads were large and polydisperse, because of the premature polymer gelation at the oil/water interface (Figure 5b) [56]. Eudragit microgels can be loaded with *Clostridium difficile* bacteriophages at a low pH and can be used for the treatment of *C. difficile* infections as an alternative to conventional antibiotic therapies. The beads are stable in the acidic environment of the stomach and release the phages at the infection site in the colon (Figure 5c). Similarly, chitosan beads can be produced by exposing droplets of acidified chitosan solution to OH^−^ ions [57]. The gelation of chitosan chains occurs as a result of the deprotonation of amine groups (NH_3_^+^) above pH 6.2–6.5.

In the second method (ii), microgels are formed by off-chip polymer crosslinking in the gelation bath. Capretto et al. [27] produced barium alginate beads in a microfluidic Y-junction by collecting the generated W/O emulsion droplets in a gelation bath containing 1.5 wt% BaCl_2_ (Figure 6). Potential issues with this technique are the accumulation of droplets at the oil/water interface as a result of the small density difference between the alginate droplets and BaCl_2_ solution, and the formation of tail-shaped beads as a result of droplet deformation during settling in the oil phase. The problems can be minimized by pouring a low viscosity oil above the BaCl_2_ solution in the gelation bath to decrease the shear force from the oil phase, and by adding glycerol to the Na-alginate solution to increase the density of the dispersed phase [27].

Two serial Y-junctions are useful for the preparation of mixed gel beads composed of two different polymers, e.g., Matrigel™ and alginate. Matrigel™ is a mixture of extracellular matrix proteins extracted from mouse sarcoma, composed of ~60% laminin, ~30% collagen IV, and ~8% entactin. Matrigel™ is liquid at 4 °C, but gels at 24–37 °C by self-assembly of laminin and collagen IV into crosslinked networks via entactin bridges. In Figure 7, a mixture of tumor cells and Matrigel in a cell culture medium at 4 °C is mixed with am alginate solution in the upstream junction, and this mixture is emulsified in mineral oil in the downstream junction. As Matrigel and alginate are delivered through separate inlet channels, a mixing ratio between the two polymers can be tuned on-chip. The droplets are crosslinked in a 4 wt% CaCl_2_ solution to form cell-laden composite beads.

In the third method (iii), a fine W/O emulsion containing aqueous CaCl_2_ droplets dispersed in corn oil is introduced through the downstream cross junction to crosslink alginate droplets formed in the upstream cross junction (Figure 8). Using two consecutive cross junctions, droplet generation is spatially separated from the crosslinking reaction to avoid clogging of the droplet-forming channel by the gel. The channel downstream of the second junction has an increased width in order to reduce the oil velocity and prevent shear-induced deformation of droplets during gelation [43]. The concentration of lipophilic surfactant used to stabilize the CaCl_2_ droplets should be sufficiently low to allow for their merging with alginate drops. The optimum concentration of Y-Glyster CRS-75 was found to be only 0.1 wt% [43]. A similar method can be used for the oxidative covalent crosslinking of modified natural polymers.

A slow gelation by CaCl_2_ nanodroplets may cause the coalescence of alginate droplets and clogging of the outlet channel. A possible solution could be to increase the length of the downstream channel, which may lead to excessive pressure buildup in the chip, particularly because of the high viscosity of the nano-emulsion. To prevent droplet coalescence, the single emulsion method can be replaced by the double emulsion method, as shown in Figure 9.

In the method shown in Figure 9, individual alginate droplets are coated by a thin layer of CaCl_2_ nano-emulsion to form core−shell droplets dispersed in the outer aqueous phase. The outer aqueous phase plays the following important roles: (i) prevents fusion of partially gelled alginate droplets; (ii) decreases viscosity in the downstream channel, which improves the chip functionality; and (iii) facilitates the separation of beads from the oil phase and minimizes cell exposure to the oil and surfactant, which increases biocompatibility of the process.

In the fourth method (vii), a self-assembling peptide (SAP) solution containing mammalian cells is emulsified in the oil phase, composed of a powdered cell culture medium dispersed in mineral oil (Figure 10a). In the downstream channel, particles of the cell culture medium collide with the droplets and become dissolved in the SAP solution. The dissolution of low molecular weight compounds, such as inorganic salts and amino acids, leads to an increase in the ionic strength within the droplets and triggers the gelation of SAPs, as shown in Figure 10b. Over 93% of the cells survived the microfluidic process and the fabricated microgels allowed for the diffusion of nutrients, as well as cell growth and differentiation [50].

#### 2.1.3. Rapid Mixing of Fluid Streams within Droplets

In this method, gelation is achieved by rapidly mixing two aqueous streams, usually a polymer solution and crosslinking solution (Figure 11a), or three aqueous streams, e.g., polymer solution, crosslinking solution, and a cell suspension (Figure 11b), immediately before droplet formation [58]. The number of cells per bead and the crosslinking density can be controlled by adjusting the flow rate ratio and composition of the inlet aqueous streams. The examples of fluid compositions and channel geometries used in this method are provided in Appendix A.

In Figure 11a, aqueous droplets comprising alginate and CaCl_2_ are formed in an immiscible continuous phase, and the gelation is achieved by chaotic advection within the droplets. Chaotic advection is caused by hydrodynamic interactions between the droplets and the channel walls. Winding collection channels provide more efficient internal fluid circulations than straight channels. The T-junction shown in Figure 11b is composed of three converging inlet channels and can be used for the encapsulation of cells within alginate microgels [60]. The time required for crosslinking (gel formation time) should be longer than the droplet formation time to prevent premature gelation and channel clogging. For constant geometry of the microfluidic channels, the gel formation time primarily depends on the concentration of the reagents, while the droplet formation time mainly depends on the fluid flowrates and channel size [59]. The gelation of alginate was also achieved by in situ mixing a solution of slow hydrolyzing acid and alginate/CaCO_3_ solution [59].

To prevent premature alginate crosslinking inside a droplet, a stream of water can be injected between CaCl_2_ and alginate solutions, as shown in Figure 12. At the upstream junction, the two side streams, namely the sodium alginate and CaCl_2_ solutions, are separated by deionized water injected through the middle channel to prevent a crosslinking reaction before the droplets are formed in the downstream junction. The droplets are solidified in the wavy reaction channel. To prevent the diffusive transport of Ca^2+^ ions through the thin layer of the middle stream, the thickness of the middle stream must be greater than the diffusional distance, x, given by $x=\sqrt{2\;D\;t}$, where D is the diffusion coefficient of Ca^2+^ and t is the contact time of the two laminar streams [61]. The concentration of the CaCl_2_ and alginate solutions plays an important role. The optimum CaCl_2_ concentration was found to be 0.5–1.0 wt% [61]. At a CaCl_2_ concentration below 0.5 wt%, droplets cannot polymerize, and the jetting regime occurs at the CaCl_2_ concentration of 2 wt% because of fast gelation.

#### 2.1.4. Merging of Polymer and Crosslinker Droplets or Injection of Continuous Stream of Crosslinking Solution into Polymer Droplets

Here, microgels are produced by injecting a crosslinking solution into the polymer droplets or by merging the polymer droplets and crosslinking solution droplets. The chip shown in Figure 13 consists of two double T-junctions for the encapsulation of glucose oxidase (GOx), horseradish peroxidase (HRP), and Amplex^®^ Red within alginate beads (T1), and for the colorimetric detection of glucose (T2). Aqueous alginic acid droplets loaded with GOx, HRP, and Amplex^®^ Red are generated at the junction J1 and are merged with a stream of CaCl_2_ solution injected from inlet A3. The fused droplets generated at the junction T1 turn to solid hydrogel particles as they move through the serpentine channel between T1 and T2. A glucose-containing sample supplied from the inlet A4 is injected into the beads at junction T2. In the presence of hydrogen peroxide, released upon enzymatic oxidation of glucose within the beads, Amplex^®^ Red reagent is transformed into highly fluorescent resorufin. The merging efficiency of double T-junctions was over 90% under optimal conditions, and was higher than the merging efficiency of single T-junctions [63].

The crosslinking strategy shown in Figure 14 is based on the fusion of alginate and CaCl_2_ droplets in a cylindrical fusion chamber. A cell-laden alginate solution and an alginate solution loaded with magnetic nanoparticles were injected through two separate inlets of the head-on junction to create a biphasic flow. Further downstream, this bicolored stream was split into Janus droplets by flow focusing with an oil phase. To inhibit the mixing of the two solutions within a droplet, the diameter of the Janus droplets was limited to 80% of the channel width [64]. The CaCl_2_ droplets were delivered to the main channel from the side channel, and their production was synchronized with the production of Janus droplets to place CaCl_2_ droplets between each two adjacent Janus droplets. The Janus droplets were merged with CaCl_2_ droplets in a cylindrical chamber to form Janus particles with a magnetic anisotropy. The magnetic halves allow for particle manipulation by a magnetic field, while the alginate halves provide an optimum microenvironment for cell growth.

A similar strategy was applied in the chip shown in Figure 15. Here, alginate and CaCl_2_ droplets formed in separate cross junctions were alternately fed to the expansion chambers where they merged and formed microgels.

As shown by others [65,66,67], in situ gelling of droplets whose dimensions are beyond the height and/or width of a microfluidic channel can be utilized to form non-spherical microgels, such as disks, rods, and threads.

By increasing the flow rate of the continuous phase or decreasing the flow rate of the dispersed phase, the shape of the beads can be changed from threads to rods to disks to spheres, reflecting different droplet volumes [65]. The resultant droplet volume depends on the competition between the viscous forces, tending to stretch the dispersed phase into a long jet, and the interfacial tension, acting in the opposite direction. Spherical beads are formed if the droplet volume, $V_{d}$, is smaller than $\pi h^{3}/6$, where h is the channel height, which is smaller than the channel width. Non-spherical beads are generated from confined droplets [66]. Plugs (rods) and threads are formed from droplets, which are confined in two directions and have a roundness, $R\;\left(=4\pi S/L^{2}\right)$ greater than unity, where S is the projected surface area of the beads and L is the projected bead perimeter [68]. For spherical and discoidal beads, $S=\pi d^{2}/4$ and L=πd, and the roundness is R=1.

The production of disk-like magnetic alginate beads loaded with cells is shown in Figure 16. The size of the beads and the number of cells encapsulated per bead can be adjusted by controlling the fluid flow rates. Disk-like beads with flat top and bottom surfaces are formed due to vertical droplet confinement in the serpentine channel, as the equivalent droplet diameter is greater than the channel height, and the channel width is sufficiently large to prevent the formation of plugs. A disk-like shape allows for the cell division process to be monitored without image distortion [67]. Different strategies used for the production of microgels by droplet merging are summarized in Appendix A.

#### 2.1.5. Competitive Ligand Exchange Crosslinking (CLEX)

This crosslinking method is based on the competition between a gelling ion (Ca^2+^) and an exchange ion (Zn^2+^) for binding sites on chelating agents (EDDA and EDTA) and a charged polymer [69]. At pH 6.7, a mixture of Zn-EDDA and alginate will not gel, nor will a mixture of Ca-EDTA and alginate, because both complexes do not dissociate at this pH (Figure 17a,b). However, upon mixing, Zn^2+^ will be exchanged between EDDA and EDTA because of their higher affinity to EDTA, compared with Ca^2+^. It will result in the release of Ca^2+^ ions (Figure 17c), which will then crosslink the alginate, because of the higher binding affinity of Ca^2+^ to alginate than EDDA (Figure 17d). At pH > 7.2, the amount of Ca^2+^ is insufficient for gelling, as the competition between alginate and EDDA for Ca^2+^ is shifted towards Ca-EDDA at such a pH. On the other hand, at pH < 6.7, the gelation is too fast because of the high concentration of crosslinking Ca^2+^ ions, which leads to clogging at the junction where the aqueous phases meet.

The chip for producing alginate beads by CLEX is comprised of the following three inlet channels: one inlet is for the carrier oil phase and two inlets are for alginate solutions containing Ca-EDTA and Zn-EDDA, respectively (Figure 17e). Alginate should be added to both aqueous streams to balance the hydrodynamic resistance in the two inlets, as well as to avoid polymer dilution after mixing. The kinetics of the ion exchange process and the amount of released Ca^2+^ ions depend on the pH and the type of chelators used, which can be used to control the gelation kinetics and gel strength [37]. The gelation time can be controlled in the range from seconds to minutes while maintaining the pH within the physiological range [37]. It ensures enhanced cell survival rates compared with the internal gelation approach, where Ca^2+^ ions are released from Ca-EDTA or solid CaCO_3_ using an acidified oil phase, which inevitably results in a pH drop well below the physiological range.

### 2.2. Covalent Crosslinking of Droplets in Microfluidic Channels

#### 2.2.1. Enzymatic Crosslinking

The mechanical properties of ionically crosslinked natural polymers, such as the elastic modulus, and the swelling ratio are unstable because of the potential loss of crosslinking ions. However, functional groups (e.g., -OH, -COOH, and -NH_2_) of natural polymers can be chemically modified to allow for their covalent crosslinking. For example, phenol containing molecules such as tyrosine and tyramine can be conjugated to alginate via carbodiimide chemistry [70] or periodate chemistry [71]. The alginate-tyramine conjugates can be crosslinked via horseradish peroxidase (HRP)-catalyzed oxidative coupling of phenol moieties in the presence of hydrogen peroxide (H_2_O_2_). Gel networks composed of covalently crosslinked polymer chains have better mechanical properties and greater chemical and thermal stability compared with ionically crosslinked polymer networks [72]. In addition, enzymatic crosslinking offers high reaction rates under physiological conditions and a “green” approach to hydrogel synthesis, including the mildness of the reaction and biocompatible catalysts [72]. Typical microgels prepared by enzymatic crosslinking of modified polysaccharides in microfluidic chips are shown in Appendix A. Most of the research has been done using alginate-tyramine, dextran-tyramine, and hyaluronic acid-tyramine conjugates and HRP/H_2_O_2_ catalysts. The same approach can be used for crosslinking polymers functionalized with resorcinol and catechol groups.

A strategy used for the encapsulation of mammalian cells within an alginate-tyramine microgel is shown in Figure 18. Alginate-tyramine (Alg-Tyr) was synthesized by conjugating tyramine to alginate in the presence of N-hydroxy sulfosuccinimide (NHS) and 1-ethyl-(dimethylaminopropyl) carbodiimide (EDC; Figure 18a). Cell-laden microgel beads were formed by injecting an aqueous solution composed of Alg-Tyr, HRP, and cells into a co-flowing stream of liquid paraffin saturated with molecularly dissolved H_2_O_2_, (Figure 18b). H_2_O_2_ penetrates inside the droplets and triggers HRP-catalyzed crosslinking of Alg-Tyr via C-C and C-O bonding of the phenol moieties [70]. H_2_O_2_ can also be supplied from a W/O nano-emulsion composed of nanodroplets of an aqueous H_2_O_2_ solution dispersed in oil [73].

Dextran-tyramine (Dex-Tyr) conjugates can be prepared by activating hydroxyl groups of dextran with p-nitrophenyl chloroformate (PNC) and reacting the obtained Dex-PNC with tyramine, as shown in Figure 19b [74]. Microfluidic encapsulation of single cells within Dex-Tyr microgels achieved by mixing the polymer, cells, and crosslinkers in situ just before droplet pinch-off is shown in Figure 19a [75,76]. One problem with this approach is in the off-centred cell encapsulation, which may cause cell escape during the subsequent manipulation. Namely, immediately after droplet pinch-off, cells take positions close to the aqueous/oil interface as a result of temporary inertial and hydrodynamic effects. As gelation occurs within milliseconds, cells become trapped in their unwanted off-centre positions [75]. The problem can be overcome by delaying the on-chip gelation of droplets, which can be achieved by the slow diffusion of H_2_O_2_ through a PDMS wall [75], rather than by direct mixing of H_2_O_2_ and polymer solution within a droplet. Tyramine-conjugated polymer can also be crosslinked within droplets through a silicone tubing submerged in a H_2_O_2_ bath [73]. Cell centring via delayed droplet crosslinking was also applied to the crosslinking of hyaluronic acid-tyramine (Hy-Tyr) conjugate and co-crosslinking of Dex-Tyr and Hy-Tyr (Appendix A).

#### 2.2.2. Polymer-Polymer Crosslinking

Hyperbranched polyglycerol (hPG) and polyethyleneglycol (PEG) can be functionalized with acrylate groups and undergo free radical co-polymerization within cell-laden droplets upon UV irradiation in the presence of a photo-initiator [77]. As photo-initiators and UV irradiation are detrimental for cell viability, further work was focused on the use of UV- and initiator-free, thiol-ene “click” reactions between dithiolated PEG macro-crosslinkers and acrylated hPG (hPG-Ac) building blocks. Microfluidic emulsification of aqueous solutions containing PEG-dithiol, hPG-Ac, and cells is shown in Figure 20. The two polymer solutions and the cell-containing medium are injected into three separate inlets of the first cross junction, where they meet and form a coflowing stream in the microchannel. In the second junction, this stream breaks up into droplets by flow focusing with a paraffin oil. Subsequent mixing of the three liquids inside the droplets led to homogenization of the droplet content and gelation reaction [78]. Another example of polymer-polymer crosslinking by click chemistry is the reaction between azide-functionalized poly(N-(2-hydroxypropyl)-methacrylamide) (PHPMA) chains and cyclooctyne-functionalized poly(N-isopropylacrylamide) (PNIPAAm) and poly(ethylene glycol) (PEG) chains [79].

Thiol-terminated PEG (PEG-dithiol) can be synthesized by the reaction between PEG-diamine and 2-iminothiolane at room temperature, as shown in Figure 21. The chemical structure of microgel particles after crosslinking is shown in the same figure.

#### 2.2.3. Photopolymerization

In this approach, droplets composed of a mixture of functional monomers and photo-initiators are exposed to UV or visible light to initiate free-radical polymerization, as shown in Appendix A. Photopolymerization offers several advantages compared with thermal or redox initiation, including short crosslinking times, ambient reaction temperatures, and high spatial and temporal reaction control [81]. The formation of biocompatible hydrogels requires the use of cytocompatible photo-initiators, such as Irgacure^®^ 2959, 1173, 819, and 651; riboflavin phosphate; camphorquinone, and eosin Y. Visible light photoinitiation is advantageous for the encapsulation of biological materials as UV radiation can cause DNA damage and accelerate tissue aging and cancer onset. Blue light photo-initiators that can be used are camphorquinone [82], eosin Y [83], and riboflavin [84,85].

Common microgels produced by monomer crosslinking with UV light are poly(N-isopropylacrylamide) (PNIPAAm) [86] and polyacrylamide (PAAm) [87]. PAAm can be synthesized through the reaction between acrylamide (monomer) and bis-acrylamide (crosslinker) in the water phase, as shown in Figure 22 [88]. PNIPAAm can be synthesized by the photopolymerization reaction between N-isopropylacrylamide (NIPAAm) and bis-acrylamide either in water [86] or in an organic solvent, such as DMSO [89].

Water soluble pre-polymers modified by the introduction of cross-linkable molecules can be used instead of monomers. The examples of such modified polymers used for microfluidic production of microgels are dextran-hydroxyethyl methacrylate (dextran-HEMA) [90], gelatin-methacryloyl (GelMA) [91,92], poly(N-isopropylacrylamide-dimethylmaleimide), (P(NIPAAm-DMMI)) [93], poly(ethylene glycol diacrylate) (PEGDA) [94], poly(ethylene glycol methyl ether acrylate) (PEGMA) [95], poly(ethylene glycol) norbornene (PEG-NB) [83], and 6-armed acrylated PEG [96].

Linear pNIPAAm chains with pendant dimethylmaleimide (DMMI) side groups can be crosslinked by the dimerization of DMMI moieties upon UV exposure in the presence of a triplet sensitizer (Figure 23). The concentration of the poly(NIPAAm-DMMI) precursor inside the droplets should be in the semi dilute non-entangled concentration regime; therefore, above the overlap concentration of polymer coils, $C^{*}$, to ensure that the polymer chains are close enough to undergo crosslinking, but below the entanglement concentration, $C_{e}$, to avoid excessive viscosity of the dispersed phase [93]. The device shown in Figure 23a is used to generate structurally homogeneous microgels. Structural microgels (core−shell microgels with Janus shells or hollow Janus shells) can be produced using the same poly(NIPAAm-DMMI) polymer labelled with different fluorescent dyes and a modified microfluidic chip with two sequential cross junctions [97].

Biocompatible poly(ethylene glycol diacrylate) (PEGDA) and poly(ethylene glycol methyl ether acrylate) (PEGMA) microgels can be generated via UV light-induced free radical polymerization of single emulsion drops generated using aqueous PEGDA or PEGMA solutions containing a suitable photo-initiator [98], as shown in Appendix A and Figure 24. PEGDA microgels can be also generated using W/O/W double emulsion droplets containing PEGDA and a photo-initiator in the inner-most drop to minimise the use of oil phase and simplify the washing steps [99].

The continuous phase can be mineral oil [95], fluorocarbon oils [98], or hydrocarbon oils [88] containing Span 80, ABIL^®^ EM 90, or fluorinated surfactants. Droplets are polymerized in a wavy downstream channel for 5 s at a UV light intensity of 70 mW/cm^2^ [95]. A jacketed wavy channel can be used to supply nitrogen under pressure through a PDMS wall into the main channel to prevent the scavenging of radicals by oxygen [98].

Natural polymers conjugated with photopolymerizable groups are attractive alternatives to synthetic hydrogels, because they can combine light polymerizable groups with inherent cell adhesion properties, because of the presence of natural cell-binding motifs, and excellent biodegradability, because of the presence of enzyme-sensitive links [100]. The example of such modified natural polymers is gelatin methacryloyl (GelMA), which is synthesized through the reaction between gelatin and methacrylic anhydride (Figure 25a) [91,100]. The conjugation of the methacryloyl moieties occurs mainly on the primary amine groups of lysine and hydroxylysine residues, but the hydroxyl groups of serine, threonine, hydroxyproline, and hydroxylysine residues are also affected [101]. GelMA microgels can be fabricated in a flow focusing microfluidic device using a dispersed phase composed of 8 wt% GelMA and 0.2–0.5 wt% Irgacure 2959 dissolved in PBS, and exposing the droplets to UV light (Figure 25b) [102,103]. Generated MelMA beads can be coated with silica hydrogel to protect encapsulated cells from oxidative stress [91].

### 2.3. Gelation of Droplets by Temperature Trigerred Sol–Gel Transition

The gelation of stimuli-responsive polymer solutions can be achieved via a sol–gel transition triggered by a temperature or pH change. Thermo-responsive hydrogels can be divided into two groups, namely: (1) upper critical solution temperature (UCST) hydrogels and (2) lower critical solution temperature (LCST) hydrogels [104]. UCST hydrogels such as gelatin and agarose are formed by cooling the polymer solution to below UCST. Cell-loaded agarose beads can be produced by flow focusing a cell-laden agarose solution at an elevated temperature [105,106] (Figure 26). Agarose beads can also be generated by in situ mixing of cell suspension and molten agarose just before droplet generation [107]. Furthermore, agarose beads can be produced by droplet generation at room temperature using ultra-low gelling agarose with a UCST < 17 °C [108,109]. Once solidified, the beads remain solid up to 56 °C and keep their integrity during handling at room temperature. Agarose droplets can be solidified off-chip in an ice bath [105] or by trapping droplets on-chip and exposing the chip to low temperatures [110].

The number of cells entrapped per droplet is dictated by the Poisson distribution [111]. Single-cell encapsulation can be achieved using a diluted cell suspension in the dispersed phase, so that ~10% of agarose droplets will contain single cells and most of the remaining droplets will be empty [106]. The entrapment of single cells within monodispersed beads ensures monoclonal cultivation and provides identical growth conditions [112]. After the incubation period, the beads are washed with a buffer solution to remove the oil phase and are sorted by fluorescence activated cell sorting (FACS) based on their fluorescence. The cells with improved properties can be recovered from the beads through the enzymatic degradation of agarose with agarose, and can be subjected to the next round of random mutagenesis.

Gelatin microgels were fabricated by injecting a 5 wt% gelatin solution at 40 °C through silicon microchannel arrays into isooctane [113]. Droplets were generated by step microfluidic emulsification and were solidified by cooling the emulsion to 25 °C. After a slow gelation at 25 °C overnight, the droplet solidification was completed at 5 °C.

## 3. Microfluidic Production of Core−Shell Microgels

### 3.1. External Gelation of Charged Polymers

Core−shell microgel particles with an oil core and hydrogel shell were produced in a four-phase glass capillary device by on-chip crosslinking of an alginate solution in core−shell double emulsion droplets [114]. In this process, an O/W_1_ emulsion composed of soybean oil droplets dispersed in a Na-alginate solution was prepared in the emulsification tube I, and was further emulsified in soybean oil in the emulsification tube II to form O_1_/W/O_2_ emulsion with core−shell droplet morphology (Figure 27). The alginate solution in the shell was solidified in the collection tube by a CaCl_2_ solution. Since double emulsion droplets are formed via a two-step process using two dripping instabilities, the diameters of inner and outer droplets can be controlled independently by adjusting the diameters of both emulsification tubes and fluid flow rates [114].

Core−shell microgels with an oil core and a gellan shell were fabricated using O/W/O double emulsion templates. On-chip crosslinking of gellan polymer chains in the middle phase was induced by calcium acetate dissolved in the outer oil phase, Figure 28 [115].

### 3.2. Internal Gelation of Charged Polymers

Core−shell microgels can be generated by internal gelation of core−shell droplets in W/W/O or O/W/O double emulsion. The crosslinking can be achieved by adding a non-charged calcium compound such as Ca-EDTA [116,117], CaCO_3_ [118], or CaCO_3_/PAG mixture [40] to the aqueous polymer solution in the middle phase. The crosslinking reaction can be triggered either by acetic acid added to the outer oil phase or by UV irradiation of the template droplets. Typical emulsion formulations are summarized in Appendix A.

The fabrication of water-core alginate-shell microgels using a W/W/O core−shell template droplets is shown in Figure 29. The alginate chains in the shell can be crosslinked by the release of Ca^2+^ from Ca-EDTA [117] or CaCO_3_ [119], triggered by the diffusion of acetic acid dissolved in the outer oil phase. By encapsulating different types of liver cells in the core and shell region, it is possible to mimic the structure of human liver and create a portable artificial liver in each hydrogel particle. In a similar way, core−shell beads consisting of a mixture of Matrigel™, collagen, and alginate in the core (solidified by temperature control) and alginate in the shell (solidified by internal gelation using CaCO_3_ and acetic acid) were produced via a W/W/O template emulsion [118].

Hollow microgel capsules can be generated by the internal gelation of core−shell droplets within an O/W/O double emulsion [116,120]. In this approach, usually Ca-EDTA or CaCO_3_ is added to the alginate solution in the middle phase, and crosslinking is initiated on-chip by organic acid dissolved in the outer oil phase [120] or off-chip by collecting the formed double emulsion droplets in an acidified oil phase [116]. Alternatively, a photo-acid generator (PAG) can be added to the middle phase to trigger crosslinking by UV light. The fabrication of capsules through the UV irradiation of template droplets containing a mixture of alginate, PAG, and CaCO_3_ in the middle phase is shown in Figure 30 (Appendix A).

### 3.3. Photocrosslinking

The oil-free, organic-solvent-free, and surfactant-free synthesis of core−shell microgels using W/W/W emulsion droplets as templates is shown in Figure 31. Aqueous solutions of dextran (DEX) and polyethylene glycol (PEG) at sufficiently high concentrations form an aqueous two-phase system (ATPS) composed of DEX-rich and PEG-rich solutions, which can be exploited in “all-aqueous” multiphase microfluidics [121]. To form droplets composed of a PEG core and a DEX shell (Figure 31), a stream of aqueous PEG solution was focused by a stream of aqueous DEX solution at the upstream junction, and the resulting thread was broken-up into droplets by another stream of PEG solution at the downstream junction. The formed core−shell droplets were exposed to UV light to crosslink DEX in the shell via the thiol-ene reaction between alkyne-functionalized dextran (DEX-GPE) and thiol-functionalized dextran (DEX-SH) chains (Figure 31b). To prevent the loss of photo-initiator from the middle phase, it was added to all phases at an equal concentration (Appendix A).

Core−shell microgel particles consisting of a polyacrylamide (PAAm) core and a poly(N-isopropylacrylamide) (PNIPAAm) shell were fabricated using solid-in-water-in-oil (S/W/O) template emulsion [123]. In this strategy, pre-synthesized PAAm microgel particles were encapsulated in a shell liquid composed of a semi dilute solution of photocrosslinkable P(NIPAAm-DMMI) chains. The shell was crosslinked by UV light in the presence of a triplet sensitizer, Figure 23b. Core−shell gelatin methacryloyl (GelMA) microgels were fabricated from core−shell template droplets consisting of a photocurable GelMA solution in the shell and a methyl cellulose solution in the core [124].

## 4. Microfluidic Production of Structured Microgels

Microfluidics can be used to fabricate structurally heterogeneous microgels, such as multi-core, multi-shell, Janus, and composite microgels [125]. Janus particles, as shown in Figure 32a, consist of three distinguishable PNIPAm polymers (colorless and tagged with different fluorescent dyes) [97]. The central PNIPAm stream forms a colorless core of droplets, while the two side PNIPAm streams tagged with different fluorescent dyes form a Janus shaped shell. The hollow microspheres shown in Figure 32b are formed from an O/W/O emulsion with a Janus-shaped middle phase. In both cases, the template droplets are crosslinked by UV-induced dimerization of dimethylmaleimide (DMMI) side groups on a polymer backbone. Janus microgels can also be generated by the Photopolymerization of phase-separated and dewetted crosslinkable polymer droplets [126].

Microfluidics can be used to produce composite microgels with a heterogeneous chemical structure. The examples of such composite microgel particles are a pH-responsive Eudragit S100 microgel encapsulated in a poly(lactic acid) (PLA) shell (Figure 33a) [127], droplet interface bilayers (DIBs) encapsulated in an alginate shell (Figure 33b) [128], distinct hydrogel beads encapsulated in a water droplet (Figure 33c), and microgel scaffolded oil-in-water-in-oil emulsion droplets (Figure 33d). Droplet interface bilayers (DIBs) can be formed via a W/O/W/O triple emulsion through the contact of aqueous droplets in an oil phase in the presence of dissolved lipids [128]. A semipermeable microgel shell allows for the structural stability of DIBs, while allowing communication with the environment.

## 5. Conclusions

Microfluidic devices can be used to produce monodispersed microgel particles with versatile chemical compositions, physical properties, and morphologies, including composite microgels with a complex internal structure composed of different solid and liquid phases. Microfluidic flow configurations traditionally used for microfluidic emulsification, such as cross, T-, Y-, and Ψ- junctions, can be combined in different ways in a manifold to generate template droplets and crosslink them into spherical and non-spherical monodispersed particles. Microgels can be loaded with cells and their spatial arrangement and number can be controlled with a high precision. Versatile internal and external ionic crosslinking methods have been developed to crosslink charged polymers without channel clogging. Covalently crosslinked microgels with mechanically improved properties compared with their ionically crosslinked counterparts can be produced by introducing moieties that can allow for enzymatic, photo-induced, and “click” chemistry crosslinking. The choice of the most appropriate continuous and disperse phase formulations, channel geometry, and crosslinking strategy is ultimately dictated by the fluid dynamics of the selected droplet generation process, the diffusion kinetics and solubility of the crosslinking agents, the kinetics of the crosslinking reaction, and the targeted applications for which the microgels are designed. For cell culture applications, the selected crosslinking process should not compromise cell viability.

## Figures and Tables

**Figure 1 molecules-26-03752-f001:**
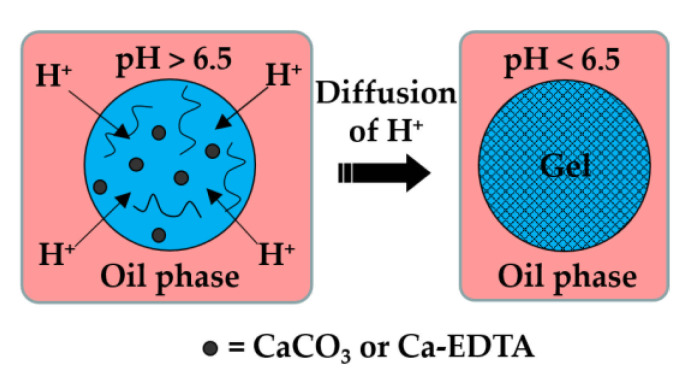
Formation of alginate beads by internal gelation. The organic acid dissolved in the oil phase dissociates at the droplet interface into acetate anions and hydrogen cations, and the pH drops below 6.5. Released H^+^ ions react with CaCO_3_ or Ca-EDTA dispersed or dissolved within the droplet, thereby releasing the Ca^2+^ ions that crosslink the polymer.

**Figure 2 molecules-26-03752-f002:**
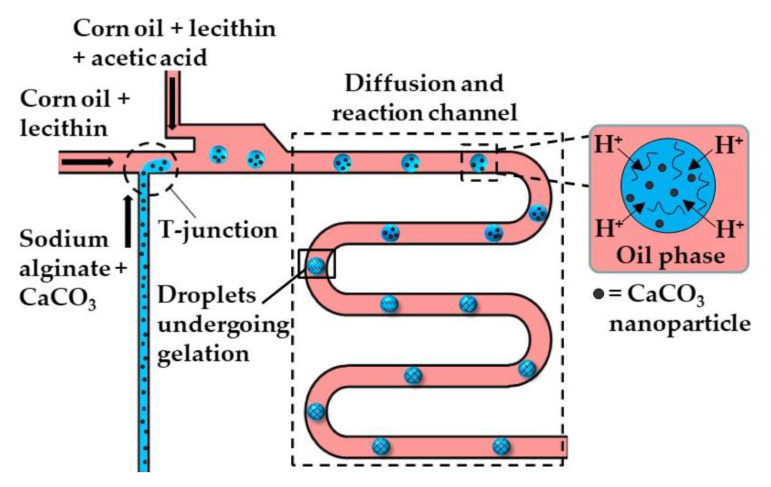
The formation of alginate beads by internal gelation. Droplets of sodium alginate solution loaded with CaCO_3_ are formed at the upstream T-junction. The resulting emulsion is mixed with acid-saturated oil delivered through the side channel, and is forced to pass through the wavy channel to allow enough time for acetic acid to diffuse to the droplets and trigger Ca^2+^ release [31].

**Figure 3 molecules-26-03752-f003:**
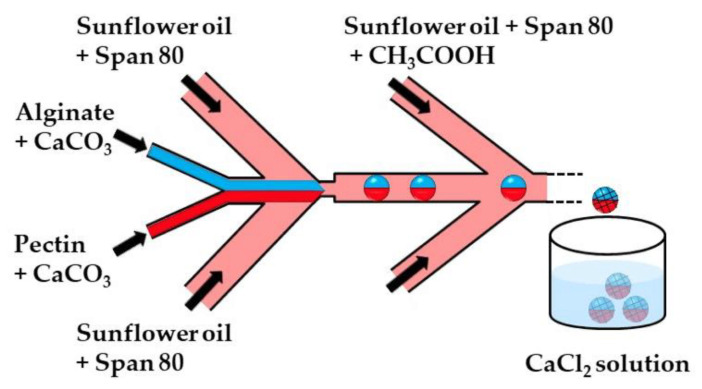
The formation of alginate/pectin Janus beads by internal gelation in the chip consisting of one upstream Y-junction and two downstream Ψ junctions. Aqueous solutions of alginate and pectin are introduced through separate inlet channels of the Y junction to form a co-axial biphasic flow. Janus droplets are formed at the upstream Ψ junction by flow focusing the coaxial jet with an oil phase, and the crosslinking reaction is initiated at the downstream Ψ junction. Because of the laminar flow, the two miscible polymer solutions remain segregated between the two Ψ junctions, enabling a Janus droplet morphology to be preserved after polymer crosslinking. The Janus beads are collected in a CaCl_2_ solution to strengthen the gel network structure [32].

**Figure 4 molecules-26-03752-f004:**
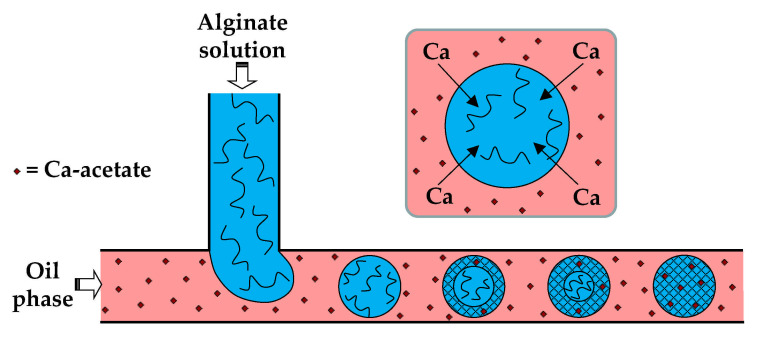
Formation of alginate beads by droplet generation in a T-junction, and the subsequent external gelation. The gelation is triggered by Ca-acetate, an oil soluble salt, which diffuses to the droplet interface and releases Ca^2+^ ions by hydrolysis, according to Equation (5) [30]. The droplet gelation occurs first in the interfacial layer, resulting in a core−shell particle morphology, and then progresses towards the particle centre.

**Figure 5 molecules-26-03752-f005:**
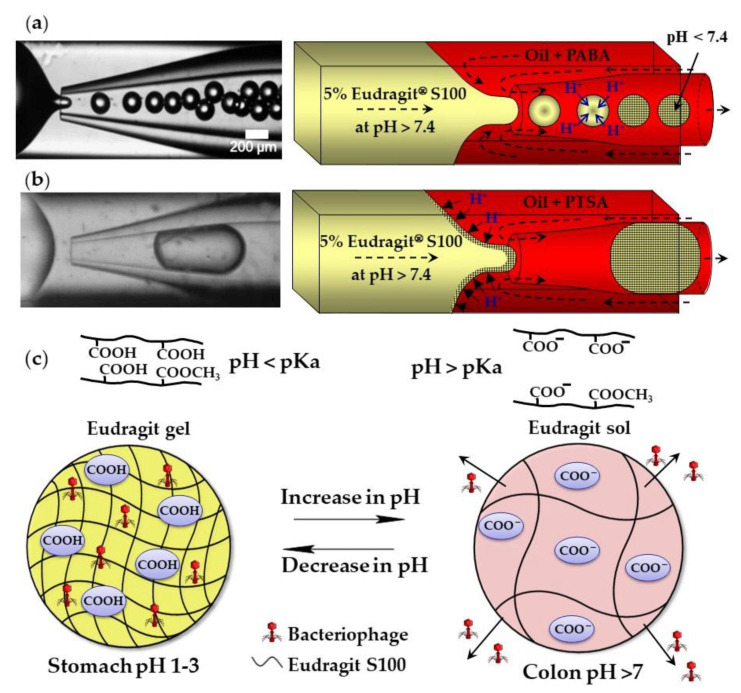
Formation of Eudragit S100 beads by external gelation of aqueous Eudragit droplets in acidified oil. (**a**) Sol–gel transition within droplets triggered by 1 wt% p-aminobenzoic acid (PABA) dissolved in mineral oil. (**b**) Premature sol–gel transition triggered by 1 wt% p-toluenesulfonic acid (PTSA) dissolved in mineral oil. (**c**) Site-specific release of bacteriophage from gel beads. The disentanglement of polymer chains occurs as a result of the electrostatic repulsion between negatively charged carboxyl groups of methacrylic acid monomer units at a pH value above the polymer pKa [56].

**Figure 6 molecules-26-03752-f006:**
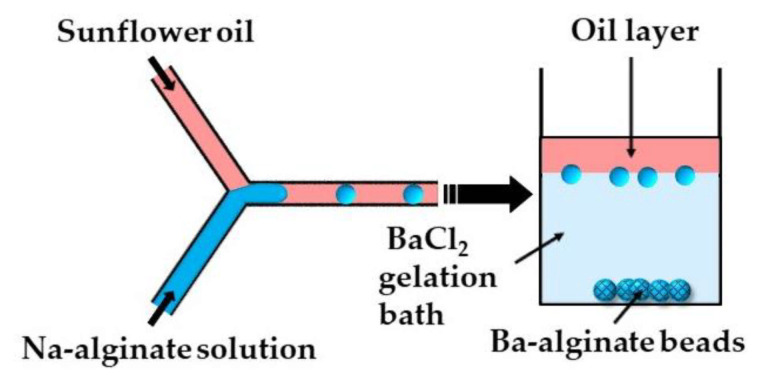
Formation of Ba-alginate beads by external gelation in a gelation bath. The continuous phase (sunflower oil) forms a top layer in the gelation bath, while alginate droplets move under gravity to a 1.5 wt% BaCl_2_ solution, where Ba^2+^ ions diffuse into droplets and cause their gelation [27].

**Figure 7 molecules-26-03752-f007:**
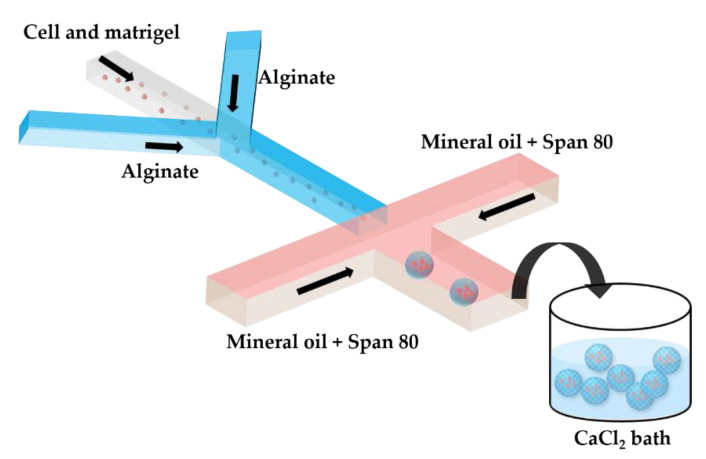
Production of alginate–Matrigel beads loaded with HeLa cells by off-chip external gelation in a gelation bath. The droplets containing HeLa cells, alginate, and Matrigel dissolved in a cell culture medium are generated in mineral oil containing 5 wt% Span 80, and are collected in a 4 wt% CaCl_2_ solution. On-line tuning of the hydrogel composition is provided by supplying the two polymers via separate inlet channels, which allows for a flexible Matrigel to alginate the mixing ratio [42].

**Figure 8 molecules-26-03752-f008:**
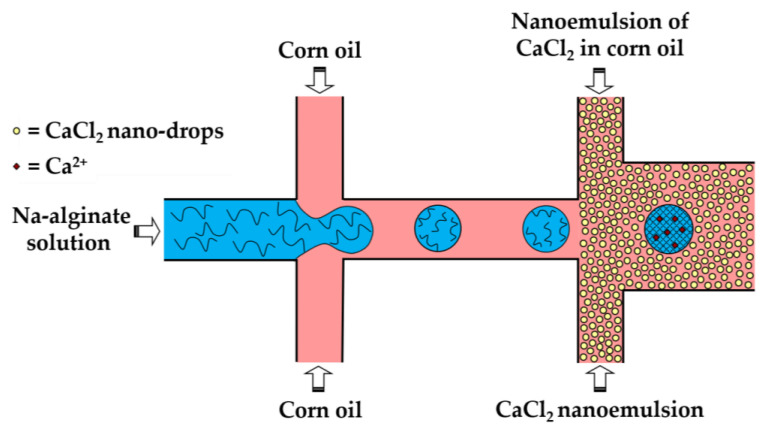
Formation of Ca-alginate beads by external gelation in a chip consisting of two serial cross-junctions using nanodroplets of an aqueous CaCl_2_ solution as the crosslinking agent [43].

**Figure 9 molecules-26-03752-f009:**
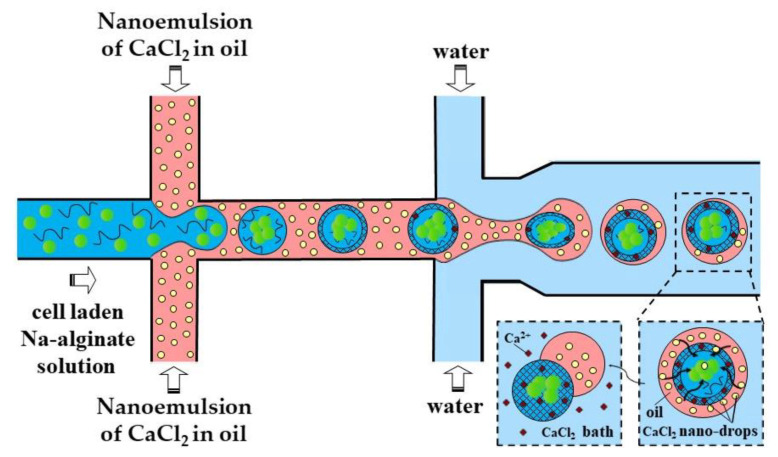
Formation of cell-laden calcium alginate beads using a two-step external gelation process within a double water/oil/water (W/O/W) emulsion. CaCl_2_ nano-droplets in the shell penetrate inside the core droplets and cause their surface gelation, followed by complete gelation in a CaCl_2_ gelation bath [44].

**Figure 10 molecules-26-03752-f010:**
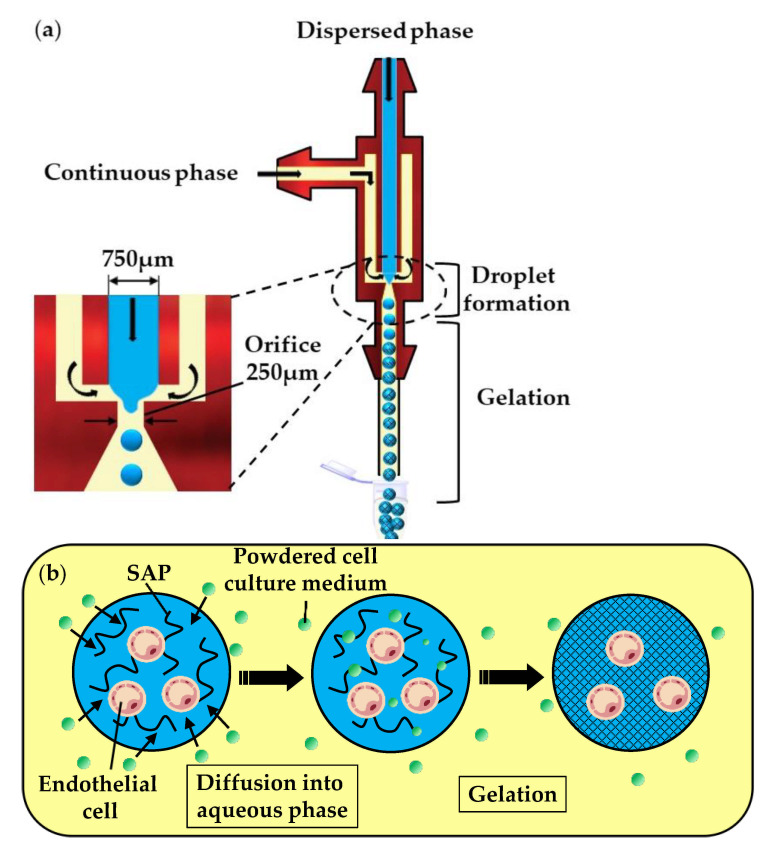
Production of self-assembling peptide (SAP) microgels loaded with bovine carotid artery endothelial cells by external gelation using a powdered cell culture medium dispersed in mineral oil. (**a**) Formation of droplets of cell-laden SAP solution. (**b**) Gelation of droplets due to an increase in the ionic strength caused by the diffusion of a powdered cell culture medium into droplets [50].

**Figure 11 molecules-26-03752-f011:**
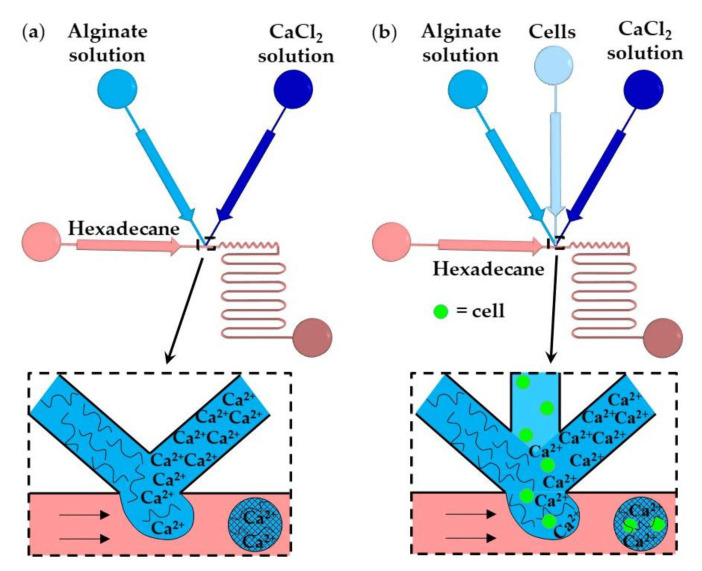
Formation of alginate beads by in-drop mixing [59]. (**a**) A microfluidic T-junction consisting of two converging inlet channels for supplying alginate and CaCl_2_ solutions. (**b**) A microfluidic T-junction for cell encapsulation within a microgel consisting of three converging inlet channels [60].

**Figure 12 molecules-26-03752-f012:**
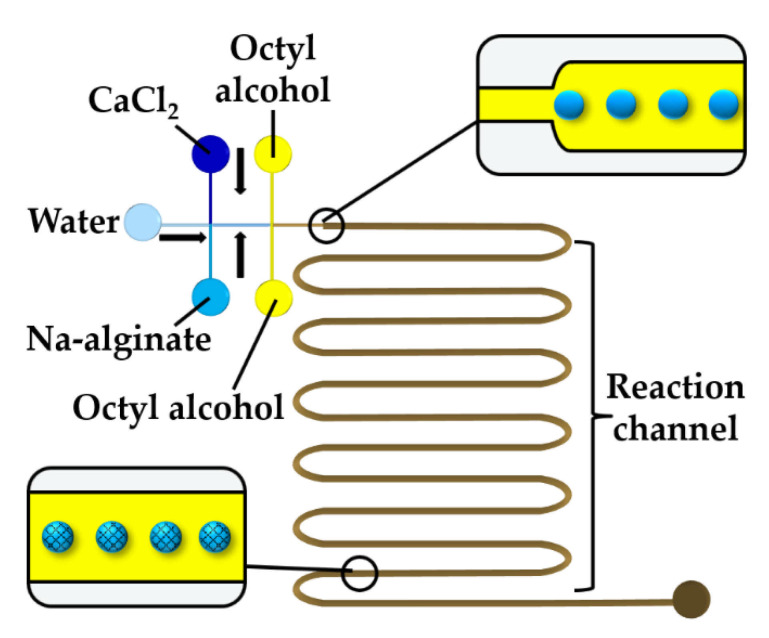
Formation of alginate beads through rapid mixing of three aqueous streams in the chip consisting of two cross junctions. The middle water stream prevents premature crosslinking [62].

**Figure 13 molecules-26-03752-f013:**
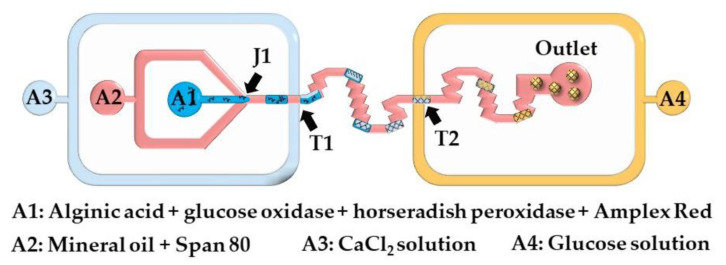
Detection of glucose in a microfluidic chip using calcium alginate beads containing glucose oxidase, horseradish peroxidase, and Amplex^®^ Red. The chip consists of one Ψ junction (J1) used for alginate droplet generation, and two double T-junctions used for injection of CaCl_2_ solution (T1) and glucose solution (T2) from the inlets A3 and A4, respectively [63].

**Figure 14 molecules-26-03752-f014:**
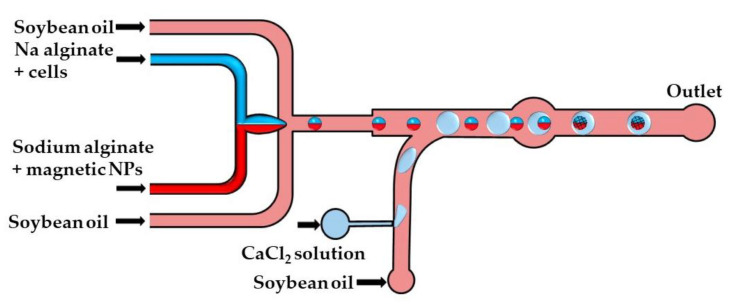
Formation of alginate/magnetic Janus beads by droplet merging in the chip consisting of head-on junction/flow focusing junction for the generation of Janus droplets, a side channel for the introduction of CaCl_2_ droplets and a fusion chamber for merging the Janus and CaCl_2_ droplets [64].

**Figure 15 molecules-26-03752-f015:**
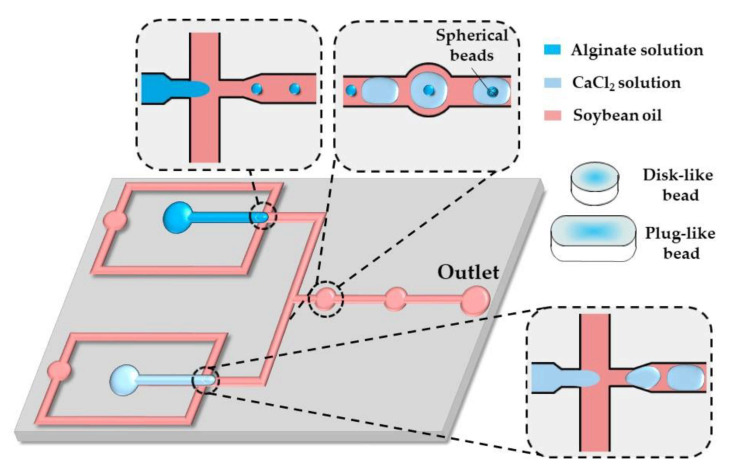
Formation of disk-like, plug-like, and spherical alginate beads in the chip consisting of two flow-focusing junctions for alternating the generation of alginate and CaCl_2_ droplets and a fusion channel with two circular expansion chambers for droplet merging. The droplet shape is modified by their confinement in one direction (disks) or two directions (plugs) [65].

**Figure 16 molecules-26-03752-f016:**
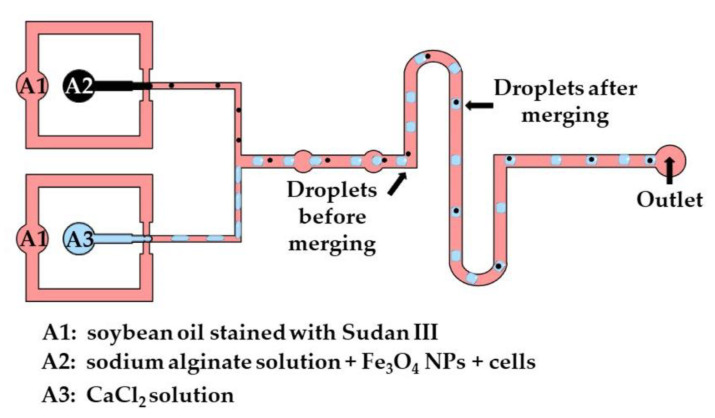
Formation of disk-like gel beads by merging the droplets of a Na-alginate solution loaded with cells and iron oxide NPs and droplets of the CaCl_2_ solution. The disk-like shape is a consequence of droplet confinement in the vertical direction before crosslinking [67].

**Figure 17 molecules-26-03752-f017:**
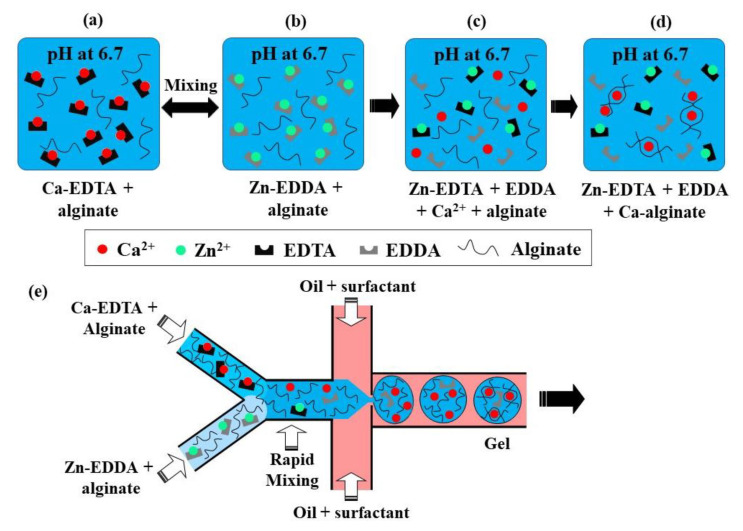
(**a–d**) Crosslinking of alginate by competitive ligand exchange. The Zn^2+^ ions are exchanged between EDDA (ethylenediaminediacetic acid) and EDTA (ethylenediaminetetraacetic acid) because of their difference in affinity, resulting in the release of Ca^2+^ ions, which crosslinks the alginate. (**e**) Formation of alginate microgels by CLEX in a microfluidic chip with three inlet streams [37].

**Figure 18 molecules-26-03752-f018:**
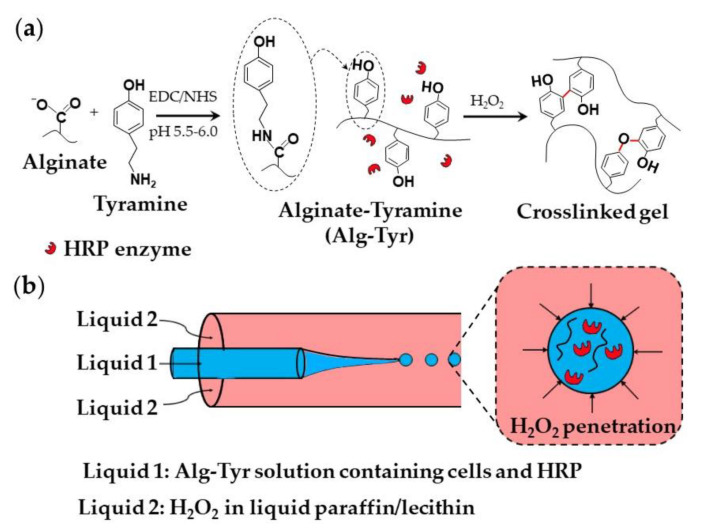
(**a**) Modification of alginate with tyramine via carbodiimide chemistry and the synthesis of alginate-tyramine gel by HRP-catalysed crosslinking of alginate-tyramine conjugate with H_2_O_2_. C-C and C-O bonds formed between phenol moieties of tyramine are shown in red. (**b**) Generation of aqueous droplets containing alginate-tyramine, cells, and HRP in a co-flowing stream of liquid paraffin saturated with hydrogen peroxide (H_2_O_2_) [70].

**Figure 19 molecules-26-03752-f019:**
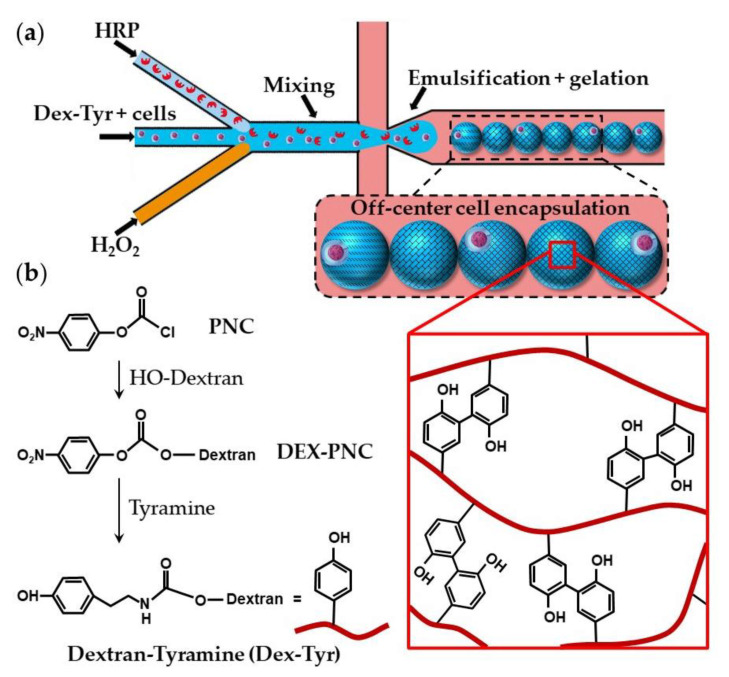
(**a**) Encapsulation of single cells using dextran-tyramine (Dex-Tyr) microgels by in situ mixing cells, polymer, and crosslinkers just before droplet pinch-off. Cells are trapped in their off-center positions because of the inertial effects and fast crosslinking reaction [75]. (**b**) Synthesis of Dex-Tyr conjugates by activating hydroxyl groups of dextran with p-nitrophenyl chloroformate (PNC). Dex-Tyr crosslinks by forming tyramine-tyramine bonds in the presence of HRP and H_2_O_2_ [74].

**Figure 20 molecules-26-03752-f020:**
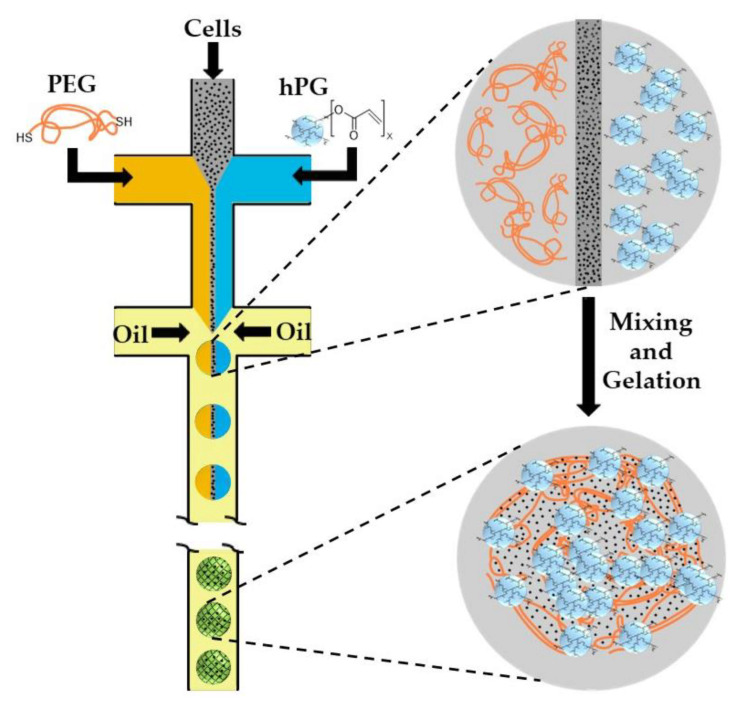
Formation of cell-laden polyethyleneglycol (PEG)-hyperbranched polyglycerol (hPG) microgel particles by thiol-ene “click” reaction. The gelation is achieved by nucleophilic Michael addition of PEG-dithiol to acrylated hPG building blocks without any initiator or UV light [78,80].

**Figure 21 molecules-26-03752-f021:**
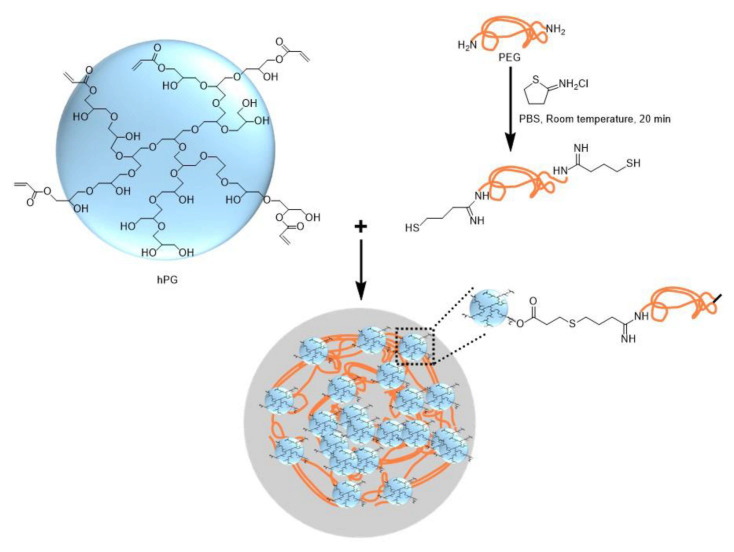
Synthesis of PEG-dithiol by reacting PEG-diamine and 2-iminothiolane and the formation of the microgel by the crosslinking of PEG-dithiol and acrylated hyperbranched polyglycerol (hPG) [78].

**Figure 22 molecules-26-03752-f022:**
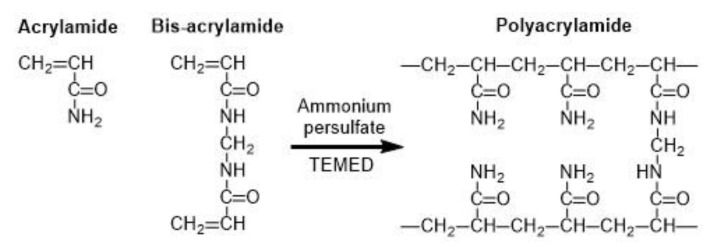
Formation of polyacrylamide (PAAm) by copolymerization of acrylamide (AAm) and bis-acrylamide (N,N’-methylene-bis-acrylamide). The persulfate free radicals initiate the polymerization reaction by converting AAm monomers to free radicals, which react with inactivated monomers. TEMED (tetramethylethylenediamine) accelerates the rate of formation of free radicals [88].

**Figure 23 molecules-26-03752-f023:**
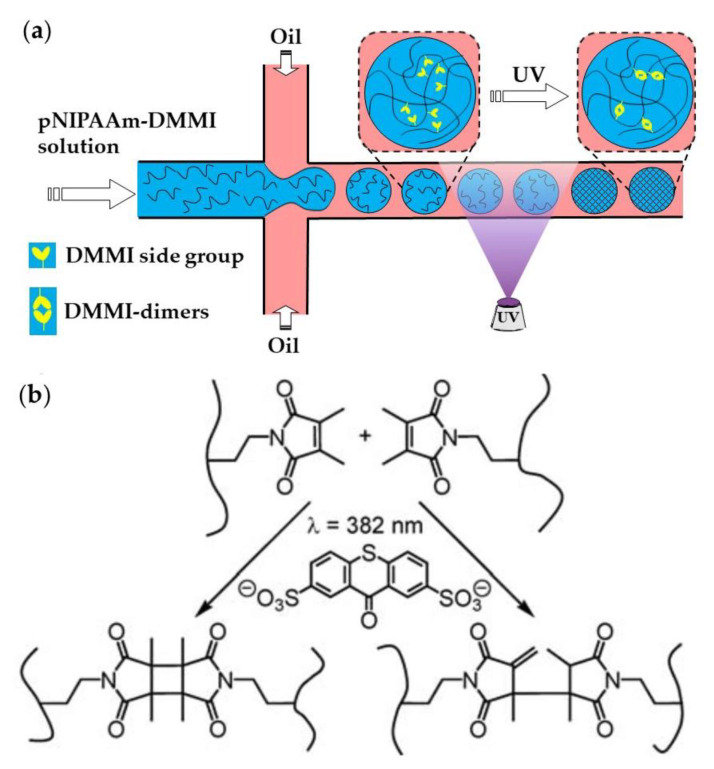
(**a**) Microfluidic synthesis of poly(*N*-isopropylacrylamide-dimethylmaleimide) microgels by Photopolymerization. (**b**) Polymer crosslinking by dimerizing side DMMI groups in the presence of thioxanthone-2,7-disulfonate (TXS). Two isomeric types of DMMI-dimers are formed [93].

**Figure 24 molecules-26-03752-f024:**
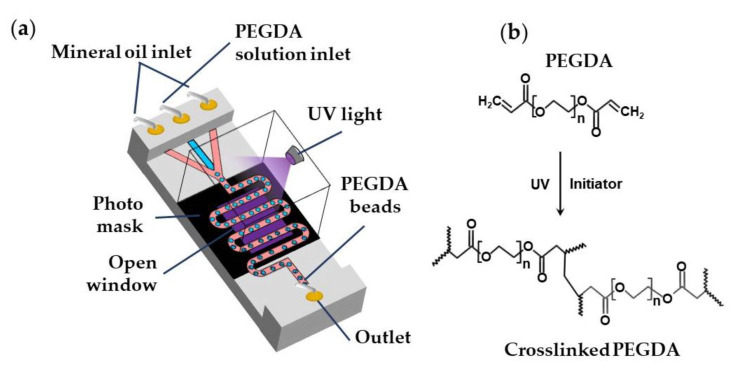
(**a**) Formation of poly(ethylene glycol diacrylate) (PEGDA) beads by the UV light-induced polymerization of aqueous PEGDA solution within droplets. A photomask can be used to expose to UV light only the wavy downstream channel, while protecting other parts of the chip [95]; (**b**) Mechanism of PEGDA photo-crosslinking reaction.

**Figure 25 molecules-26-03752-f025:**
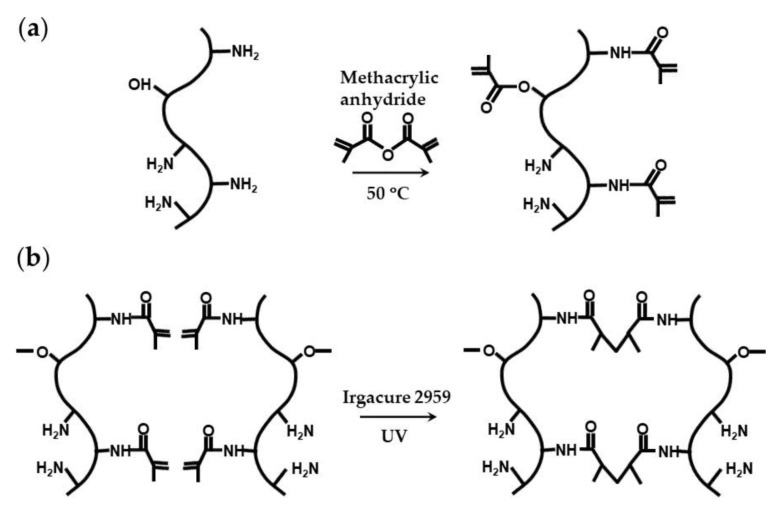
Synthesis and photocrosslinking of gelatin methacryloyl (GelMA): (**a**) Conjugation of methacryloyl groups to gelatin via primary amine and hydroxyl groups. (**b**) Photocrosslinking of GelMA through a Micheal-type addition reaction in the presence of photo-initiator [91,100].

**Figure 26 molecules-26-03752-f026:**
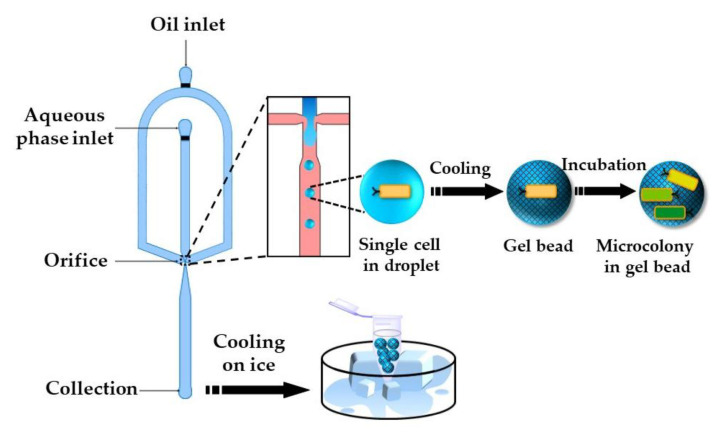
Encapsulation of single *E. coli* cells within agarose beads by temperature-controlled emulsification of an agarose solution in a flow-focusing device followed by cooling. The entrapped cells grow into monoclonal microcolonies inside the beads and can be sorted by fluorescence activated cell sorting (FACS) [105].

**Figure 27 molecules-26-03752-f027:**
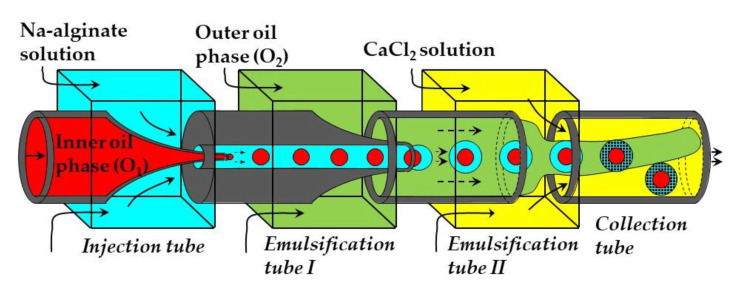
Co-flow glass capillary device for fabrication of oil-core alginate-shell microgels via on-chip gelation of shell solution within O_1_/W/O_2_ emulsion droplets. The droplets sink to the bottom of the collection tube due to density difference between the formed beads and the oil phase [114].

**Figure 28 molecules-26-03752-f028:**
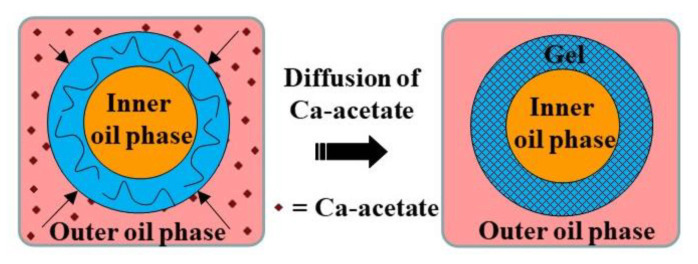
Fabrication of core−shell microgels with oil core and gel shell by external gelation of middle phase of O/W/O emulsion droplets. The gelation was triggered by diffusion of calcium acetate dissolved in the outer oil phase to the external droplet interface [115].

**Figure 29 molecules-26-03752-f029:**
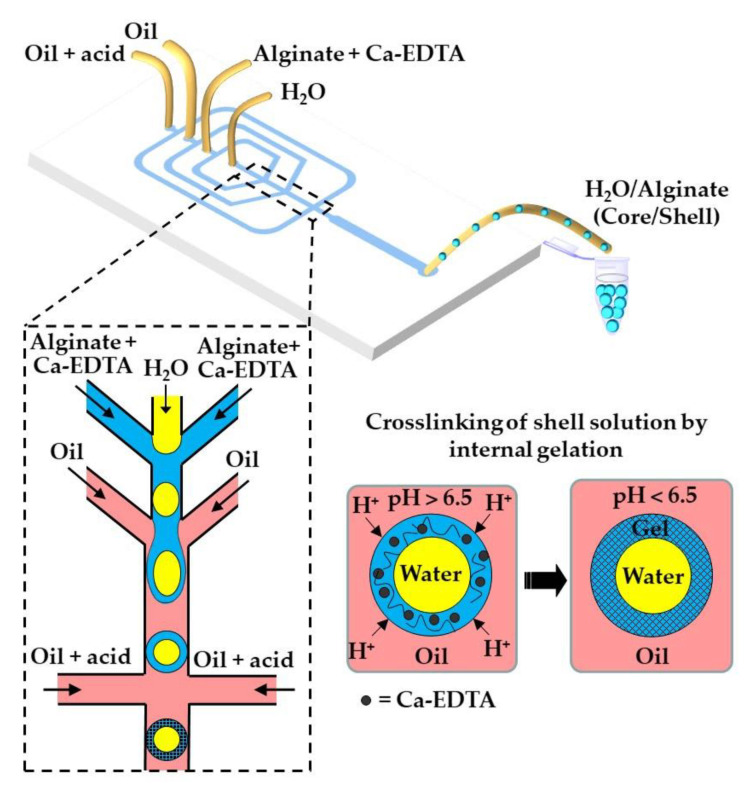
Fabrication of core−shell microgels consisting of an aqueous core and an alginate shell via W/W/O double emulsion template droplets. The alginate in the shell is crosslinked by the in situ triggered release of Ca^2+^ from Ca-EDTA. Different types of liver cells can be encapsulated in the core and shell regions to create an artificial liver on a microgel [117].

**Figure 30 molecules-26-03752-f030:**
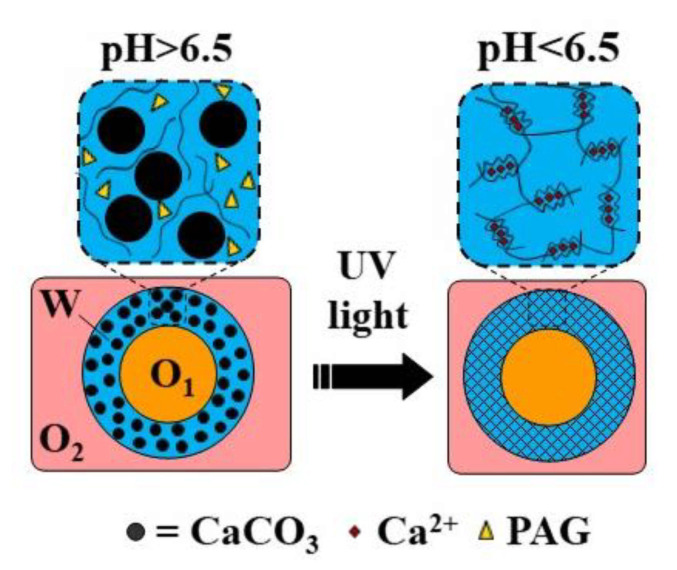
Formation of alginate capsules with an oil core through the internal gelation of the middle phase of O/W/O emulsion droplets containing CaCO_3_ and a photo-acid generator (PAG). The crosslinking is triggered by UV irradiation, which releases acid from PAG and dissolves CaCO_3_ [40].

**Figure 31 molecules-26-03752-f031:**
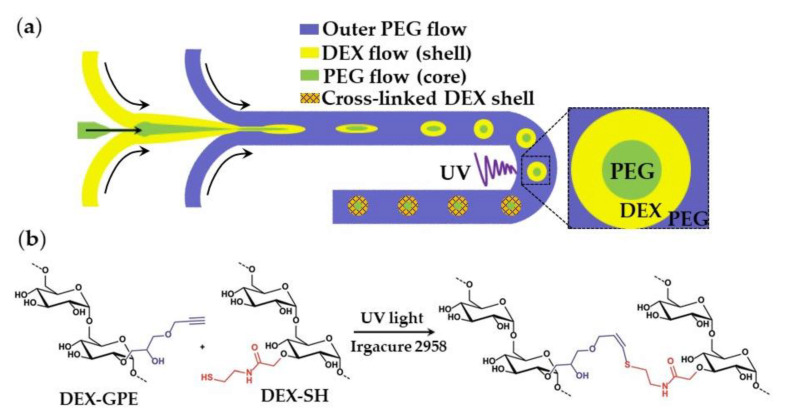
(**a**) Formation of microgels with a liquid PEG core and a solid DEX shell by photocrosslinking in a W/W/W double emulsion; (**b**) Mechanism of thiol-ene crosslinking of alkyne-functionalized dextran (DEX-GPE) and thiol-functionalized dextran (DEX-TH) in the shell [122].

**Figure 32 molecules-26-03752-f032:**
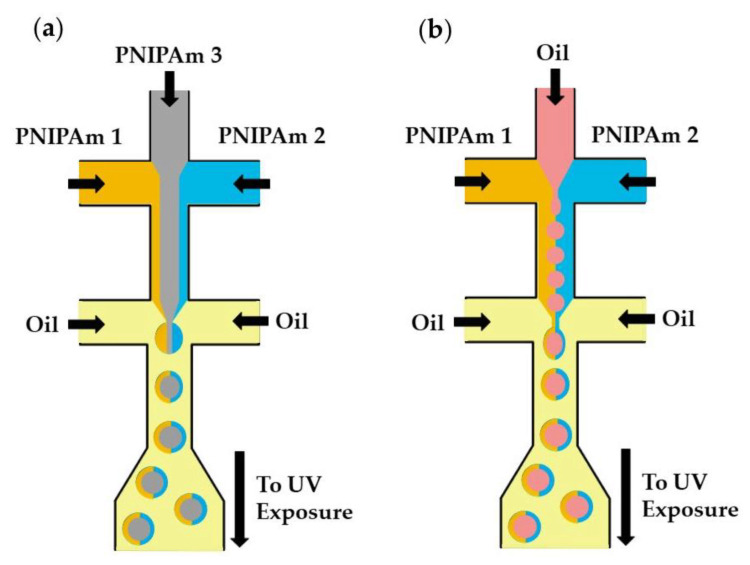
Fabrication of poly(N-isopropylacrylamide) (PNIPAm)-based Janus microgels. (**a**) Solid Janus microgels produced from three distinguishable PNIPAm streams. (**b**) Hollow Janus microgels produced from two distinguishable PNIPAm streams and one inert oil stream [97].

**Figure 33 molecules-26-03752-f033:**
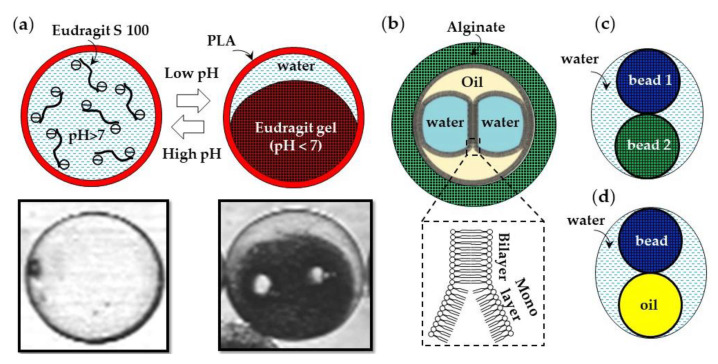
Composite microgels prepared in microfluidic devices. (**a**) pH-responsive Eudragit S100 microgel encapsulated within a Nile Red-labelled poly(lactic acid) (PLA) shell. At pH > 7, the core is a colorless liquid due to hydrophilic character of charged Eudragit chains. At pH < 7, the core is a red-colored solid gel due to diffusion of Nile red from the shell. A small amount of water was expelled from the gel after the sol–gel transition [127]. (**b**) Droplet interface bilayers (DIBs) formed in an oil phase and encapsulated within an alginate shell [128]. (**c**) Gel beads from two distinct polymers encapsulated in a single water drop using a microfluidic particle zipper [129]. (**d**) A single gel bead and oil droplet encapsulated within a larger water droplet [130].

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
