# Peer review of "Crosslinking Strategies for the Microfluidic Production of Microgels"

_molecules, 2021, doi:10.3390/molecules26123752_

Round 1

Reviewer 1 Report

This review is very well written. The majority of the microfluidic approaches used to synthesize microgels of different chemistries is carefully detailed.  

I really enjoyed reading this review and appreciated that the authors have systematically accompained the text with detailed sketches of the devices. This made the review easy-to-read. 

I recommend to publish this review in the present form.

Minor:

Line 540: ...using the same the poly(NIPAAm-DMMI) polymer...

should be

...using the same poly(NIPAAm-DMMI) polymer.... 

Author Response

Dear Reviewer,

Thank you very much for your comments.

In the revised manuscript, "the poly(NIPAAm-DMMI) polymer" was replaced by "poly(NIPAAm-DMMI) polymer".  

Reviewer 2 Report

This review paper introduces recent developments of various microfluidic-based methods for micro gel production. Microgels are mainly divided into a homogeneous or core-shell structure, and the microfluidic formation of microgels provides the ability to precisely tune chemical composition, size, shape, surface morphology, and internal structure of microgels. In this paper, gel formation mechanisms and microgel production mechanisms using microfluidic devices were well described by showing detailed figures. The reviewer suggests publishing this review paper after minor revisions as below.

  1. The authors should describe the mechanism of microdroplet formation using microfluidic channels. Although the authors slightly explained droplet microchannels in conclusion (line 732), in the main text, detailed mechanisms of microdroplet formation in microchannels were not described. An additional section of “microdroplet formation in microfluidic channels” would help readers understand the advantages, characteristics, and mechanisms of droplet microchannels.

  1. The figure number should be revised from figure 29.

  1. Figure 7 was not cited in the main text.

Author Response

Dear Reviewer,

Thank you very much for your useful suggestions. 

1. The authors should describe the mechanism of microdroplet formation using microfluidic channels.

In the revised manuscript, we have added several review papers in which the mechanisms of droplet generation are described in detail.  

2. The figure number should be revised from figure 29.

In the revised manuscript, the figure numbers are revised from figure 29 onwards. 

3. Figure 7 was not cited in the main text.

The main text was checked but Figure 7 was cited in the main text.